

# A fully automated approach involving neuroimaging and deep learning for Parkinson's disease detection and severity prediction

Çağatay Berke Erdaş and Emre Sümer

Department of Computer Engineering/Faculty of Engineering, Başkent University, Ankara, Türkiye

## ABSTRACT

Three-dimensional magnetic resonance imaging has been proved to detect and predict the severity of progressive neurodegenerative disorders such as Parkinson's disease. The application of pre-processing with neuroimaging methods plays a vital role in post-processing for these problems. The development of technology over the years has enabled the use of deep learning methods such as convolutional neural networks (CNN) on magnetic resonance imaging (MRI) . In this study, the detection of Parkinson's disease and the prediction of disease severity were studied with 2D and 3D CNN using T1-weighted MRIs that were pre-processed with FLIRT image registration and BET non-brain tissue scraper. For 2D CNN, the median slices of the MR images in the sagittal, coronal, and axial planes were used separately and in combination. In addition, the whole brain for 3D CNN has been downsized. Considering the performance of the proposed methods, the highest results achieved for detecting Parkinson's disease were measured as 0.9620, 0.9452, 0.9407, and 0.9536 for Accuracy, F1 score, precision, and Recall, respectively. The highest result achieved for estimating the severity of Parkinson's disease was that 3D CNN was fed three times with a downsized whole MRI, which were measured for R, and $R^2$ as 0.9150 and 0.8372, respectively. When the results obtained with the methods suggested within the scope of the study were examined, it was observed that the applied methods yielded promising performance.

# INTRODUCTION

Parkinsonism, as a clinical syndrome, is characterized by tremors, rigidity, bradykinesia, akinesia, and postural abnormalities (*Niemann & Jankovic, 2019*; *Heim et al., 2021*). Parkinson's disease (PD), on the other hand, is one of the most frequent variants of Parkinsonism syndrome, and it is the most common condition among the movement disorder group diseases after essential tremor (*Algarni & Fasano, 2018*). PD is distinguished by its specific pathology, clinical picture, and distinct response to dopaminergic therapy. As this situation makes treatability a critical criterion for diagnosing PD, it also turns the correct diagnosis of PD into an essential condition of treatment success. Because PD is a neurodegenerative disease for which pharmacological treatment is most successful,

Corresponding author
Çağatay Berke Erdaş,
berkeerdas@gmail.com

accurate estimation of disease grade, as well as disease diagnosis, will determine the dosage at which the patient is exposed to the drug. This will improve the patient's treatment and quality of life (*Calne, 2005*).

Typically, PD is a disease of middle and old age; it starts at the average age of 50–60 years and progresses gradually for approximately 10–20 years. PD is the prototype picture for hypokinetic diseases and is characterized by Parkinsonian motor signs.

The motor signs of PD described above arise because of the damage and loss (degeneration) of a small part of the nerve cells responsible for movements in the brain. These cells secrete a chemical substance called dopamine that sends information from one nerve cell to another. If sufficient dopamine cannot be produced in the brain, the movement and posture functions are affected, and the symptoms of Parkinson's disease occur. This degeneration in the brain is a degenerative process that affects the basal ganglia, primarily substantia nigra, and other brain stem-pigmented neurons and constitutes 80% of all Parkinsonism cases.

Although the gold standard in diagnosing PD is still neurological examination, with the developing technology, decision support systems have been developed to assist physicians using various computer-integrated methods. In this context, degeneration in the brain can be detected by examining the brain images obtained using neuroimaging techniques in computer science. *Pyatigorskaya et al. (2018)* aimed to compare the diagnostic efficaciousness of DTI, NM-sensitive imaging, and DNH and find the appropriate combination of meters to perceive substantia nigra alterations in PD. *Cigdem, Beheshti & Demirel (2018)* investigated the f-contrast hypothesis, which assesses the differences between two groups without any direction limitations. Their experimental results indicate that using f-contrast improves the classification accuracy significantly. In addition to pure neuroimaging-based studies, there are studies in the literature supported by machine learning approaches (*Morales et al., 2013*; *Nair et al., 2013*; *Salvatore et al., 2014*; *Haller et al., 2012*; *Duchesne, Rolland & Vérin, 2009*; *Acton & Newberg, 2006*). *Salvatore et al. (2014)* worked on the differential diagnosis problem for progressive supranuclear palsy (PSP), PD, and healthy control (HC) by using magnetic resonance imaging (MRI) scans of 84 patients. They used principal component analysis (PCA) as a feature extractor and support vector machine (SVM) as a classification technique. *Long et al. (2012)* studied functional magnetic resonance imaging (rs-fMRI) and structural images to classify early PD patients and the control group. They developed a classification method for these areas by extracting regional functional connectivity strength, regional homogeneity, and amplitude of low-frequency fluctuations over rs-fMRI, and white and gray matter from structural images. Most machine learning techniques rely on hand-crafted features, with the most important features being manually selected. Different dimensionality reduction techniques are typically used to remove less critical features.

In this study, a solution was sought for PD detection and PD severity estimation problems with 2D and 3D convolutional neural network (CNN) deep learning methods using T1-weighted (T1w) MRIs, which are automatically pre-processed without any human intervention. In this way, the human factor has been eliminated, thus preventing possible wrong actions and ensuring objectivity for all transactions. Also, more MRIs can

be processed by automating the relevant process, thereby expanding the sample space in which the proposed methods are tested. Moreover, 2D and 3D CNN methods were fed using slice-based and whole-brain-based approaches. Detailed explanations of these approaches are provided in the general overview section. In the problem of predicting the severity of Parkinson's disease, the evaluation results obtained with the Hoehn & Yahr scale (*Siciliano et al., 2017*) were used as an objective measure.

## MATERIALS & METHODS

### General overview

In this study, a total of 1130 T1w MR images collected from 259 healthy individuals who were members of the control group, and 871 Parkinson's patients were used to detect Parkinson's disease and to predict the severity of the disease. In this context, MRIs of different sizes obtained from different devices were subjected to fully automatic neuroimaging pre-processing to transform them into a uniform model and to remove unnecessary non-brain tissue. To find solutions to Parkinson's disease detection and disease severity prediction problems, T1w Brain MRIs, which became uniform and purified from non-brain tissue with the pre-processing stage, fed two main approaches called (i) median slices and (ii) whole brain. In the median slices approach, the slices belonging to the Sagittal, Coronal, and Axial planes were resized in $224 \times 224$ and fed the 2D CNN deep learning method separately. Besides, these extracted and resized slices were placed in the channels in the order mentioned above, and a new image was obtained, which is $224 \times 224 \times 3$ in size. This image containing information about each plane was processed using 2D CNN deep learning methods, and a solution was sought. The whole-brain approach's main purpose is to process the entire brain with 3D CNN architecture after pre-processing. However, due to the numerical excess of MR images and the size of each MRI, the size of the processed data has yet to make it possible to process an effective 3D CNN deep learning method with today's technology. For this reason, the pre-processed MRIs with the size of $182 \times 218 \times 182$ have been reduced to $61 \times 73 \times 61$ and $46 \times 55 \times 46$. In other words, processed MR images were downsized about 3 and 4 times. At this stage, a 3D CNN method was used in line with the PD detection and PD severity estimation problems by feeding downsized MR images. Figure 1 includes the illustration of the proposed general structure.

### Convolutional neural network

A convolutional neural network (CNN) architecture is a multi-layered feed-forward neural network built by stacking many hidden layers on top of each other in sequence. CNN can learn hierarchical features thanks to their sequential design (*Beyaz, Açıcı & Sümer, 2020*). Convolutional layers are typically followed by activation layers, with some by pooling layers. Although the CNN architectures using these layers were originally designed as 2D, they are now adapted to 1D and 3D. Within the scope of this study, slices extracted from the planes fed 2D CNN, while 3D CNN processed the downsized T1w MR images. The 2D CNN architecture was inspired by the AlexNet architecture. It consists of 5 convolutional layers and three fully connected layers. More information about this architecture can be found in *Krizhevsky, Sutskever & Hinton (2017)*. While a more complex architecture is

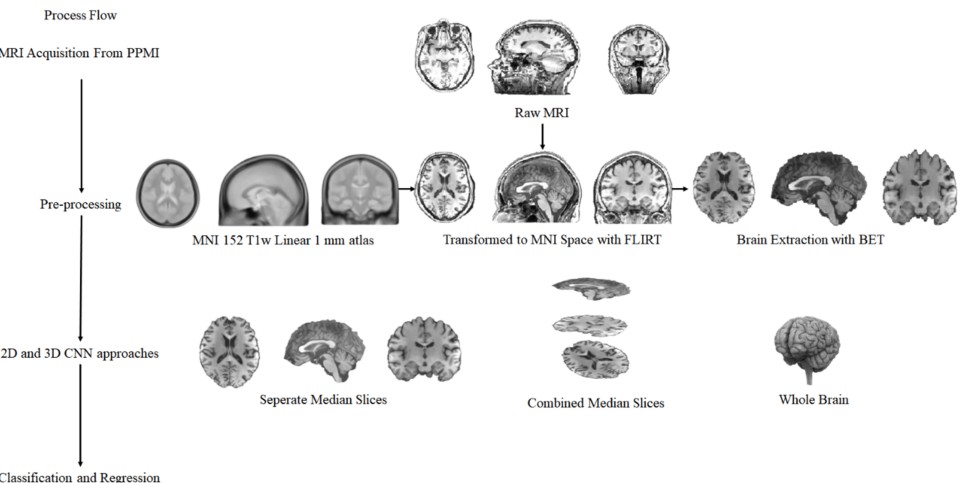

**Figure 1** The complete process of general structure.

used in 2D CNN, a relatively simple structure is preferred in 3D CNN due to the size of the data. There are four convolutional layers and three fully connected layers in this 3D CNN architecture used. While the kernel was determined as 3 in all convolutional layers, the number of filters was 64. After these layers utilize ReLU as the activation function, the max-pooling layer has been added so that the pool size is 2. After the convolutional layers, a flatten layer was applied to fully connected layers. While the number of neurons in the first two fully connected layers is 1,000, this number differs from the classification or regression problem in the last fully connected layers. For both architectures, the output of the network varies for the classification and regression problem. In the classification, a softmax layer and a dense structure with two neurons for output were used after the last fully connected layer. At the same time, there is no softmax layer in regression, and there is a single neuron in the integrated dense layer for output. This architecture was used when training both 2D and 3D CNN from scratch.

## Data acquisition and pre-processing
### Data acquisition
Data for this research were obtained from the Parkinson's Progression Markers Initiative (PPMI) database (https://www.ppmi-info.org/access-data-specimens/data). The PPMI neuroimaging database is considered an international landmark and multicentre study for biomarker research responsible for Parkinson's disease progression. There are many types and numbers of MR images under PPMI. The MR images required for this study were selected according to the criteria. The search criterion aims to find a T1w structural MRI with PD and Control constraints on all runs in PPMI. Search criteria are independent of constraints such as age, gender, and scanning device. A total of 1,130 MRI scans were chosen after applying the filter based on the imaging protocol. There were 259 healthy individuals and 871 Parkinson's patients. The scans used in the study belong to people $62.64 \pm 9.9$ years old.

## Data pre-processing

The study's dataset came from the PPMI database; since PPMI is a multicentre initiative, the imaging scans acquired in the study contained temporal and spatial variations. To solve this problem and retain a consistent modality for all MRI scans, all scans had to be in the same layout, such as the Montreal Neurological Institute (MNI) (*Fonov et al., 2011*). As a result, an image registration technique was performed to integrate the PPMI MRI data obtained from various centres worldwide into a fixed coordinate system. Image registration is a technique for finding orientation parameters and coordinates on a fixed image (atlas) such that an unseen or unknown image has been aligned appropriately to the fixed image. In this study, the MRI scans collected from the PPMI were the source image, while the atlas, such as MNI, was considered to be the target. The registration process was carried out on MRIs collected from the PPMI database, with MNI 152 T1w Linear one mm atlas (*Mazziotta, 2002*) reference by developing a fully automated Python code base on the FLIRT registration tool (*Smith et al., 2004*) belonging to the FMRIB Software Library (FSL).

Once the image registration is completed and all MRIs are aligned, removing unwanted tissues such as bone, skin, fat, air, and anatomical structures such as the neck, upper spinal cord, eyes, and mouth in MRIs increases the performance of the method to be applied. These structures are not needed in studies based on the degeneration of PD in the brain. Therefore, it is best to extract only the brain-related tissue. In this context, FSL's BET (*Smith, 2002*) method was applied with a 0.5 threshold to eliminate unnecessary structures and extract the brain. It was operated without human factors for all MRIs in the data set.

# RESULTS

## Performance evaluation

In all experiments conducted within the scope of this study, the k-fold cross-validation technique was used. In the k-fold cross-validation technique, the data set is divided into k parts, each with an equal number of randomly determined samples. While each part is separated for testing, the training process is carried out with the remaining parts. This process continues until each part is used for testing. In this way, it is ensured that each sample can be used independently for both testing and training. The k value for this study was determined to be 10. To test the performance of the classification methods for the Parkinson's disease detection problem, Accuracy, F1 score, Precision, and Recall metrics were used, while the correlation coefficient (R), coefficient of determination ($R^2$ score), Mean Absolute Error (MAE), Median Absolute Error (MedAE), Mean Squared Error (MSE), and Root Mean Squared Error (RMSE) were used to test the performance of the regression methods developed for the prediction of the disease severity.

## Empirical results and findings

The studies conducted within the scope of this study were divided into two groups such as Parkinson's disease detection, which is a classification problem, and Parkinson's disease severity prediction, which can be considered a regression problem.

**Table 1  Classification results obtained for Parkinson's disease detection.**

|  | Accuracy | $F_1$ Score | Precision | Recall |
|---|---|---|---|---|
| Median Slices | 0.9620 | 0.9452 | 0.9407 | 0.9536 |
| Axial Median Slice | 0.9319 | 0.8992 | 0.9162 | 0.8825 |
| Coronal Median Slice | 0.9381 | 0.9074 | 0.9341 | 0.8866 |
| Sagittal Median Slice | 0.9354 | 0.9036 | 0.9292 | 0.8835 |
| All Brain Downsized × 3 | 0.9558 | 0.9046 | 0.8943 | 0.9151 |
| All Brain Downsized × 4 | 0.9549 | 0.8953 | 0.8417 | 0.9561 |

Table 1 contains the results of the proposed PD detection classification problem methods. Accordingly, the best performance was obtained with 2D CNN in the median slices methods. It is a technique based on combining the median slices extracted from Sagittal, Coronal, and Axial planes, and the results were obtained as 0.9620, 0.9452, 0.9407, and 0.9536 for Accuracy, F1 score, Precision, and Recall, respectively. When looking at the results obtained when the median slices extracted from the Sagittal, Coronal, and Axial planes are used alone, it can be observed that the classification performance is close to each other, and no plane stands out. The results obtained with the 3D CNN method in which the whole brain is used, the performance results are very close, but the results after reducing the original size to one-third are better. Hence, the classification performance of the All Brain Downsized × 3 methods was measured as 0.9558, 0.9046, 0.8943, and 0.9151 for Accuracy, F1 score, Precision, and Recall, respectively. When the general table is examined, it is evident that the best result achieved for classification is the Median Slices method.

Table 2 contains the results of the proposed methods for disease severity prediction. Therefore, the results obtained by combining all planes among Median slice methods provided the best output. For this method, the results were computed for R, $R^2$, MAE, MedAE, MSE, and RMSE as 0.915, 0.8372, 0.1387, 0.0168, 0.1287, and 0.3587, respectively. In addition, the results of 2D CNN using only the sagittal plane surpassed those obtained from other planes and were almost as successful as combinations of medians. When the results obtained in the regression study on the whole brain with 3D CNN are examined, it can be observed that the results of All Brain Downsized × 3 surpass the other method. The results obtained with the All Brain Downsized × 3 methods were measured as 0.9286, 0.8622, 0.1576, 0.0389, 0.1089, 0.33 for R, $R^2$, MAE, MedAE, MSE, and RMSE, respectively. Also, it can be observed that 3D CNN results using All Brain Downsized × 3 are the best for disease severity prediction.

To compare the success of the proposed hybrid method, the relevant dataset was pre-processed in its raw form, FLIRT and BET, and subsequently processed with artificial intelligence algorithms. In this context, learning could not be observed in the classification and regression models trained on raw images. Although the results obtained with FLIRT or BET pre-process methods showed promised performance, both of them could achieve a different level of results than the proposed method in their respective sub-problems. Figures 2 and 3 show the averages of classification and regression results from each of the Median Slices, Axial Median Slice, Coronal Median Slice, Sagittal Median Slice, All Brain

**Table 2   Regression results obtained for prediction of Parkinson's disease severity.**

|  | R | R² | MAE | MedAE | MSE | RMSE |
|---|---|---|---|---|---|---|
| Median Slices | 0.915 | 0.8372 | 0.1387 | 0.0168 | 0.1287 | 0.3587 |
| Axial Median Slice | 0.8734 | 0.7629 | 0.196 | 0.0241 | 0.1875 | 0.433 |
| Coronal Median Slice | 0.8915 | 0.7948 | 0.1868 | 0. 0253 | 0.1623 | 0.4028 |
| Sagittal Median Slice | 0.9148 | 0.8368 | 0.1608 | 0.0217 | 0.1291 | 0.3592 |
| All Brain Downsized × 3 | 0.9286 | 0.8622 | 0.1576 | 0.0389 | 0.1089 | 0.33 |
| All Brain Downsized × 4 | 0.8774 | 0.7698 | 0.2252 | 0.0557 | 0.1821 | 0.4267 |

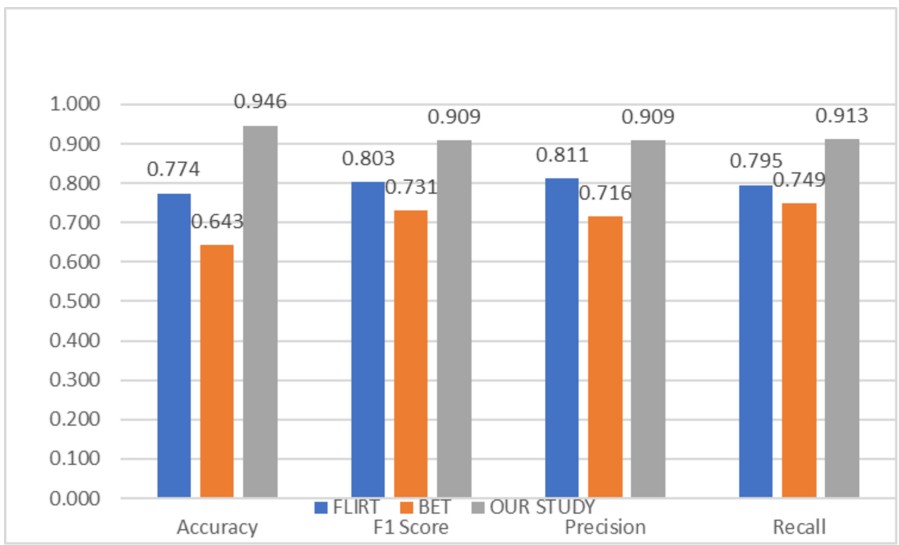

**Figure 2   Average classification performance achieved by preprocessing techniques.**

Downsized × 3, All Brain Downsized × 4 techniques using FLIRT, BET, and the proposed method, respectively.

In Fig. 2, the average of the classification performances obtained over the sub-problems with three different methods (FLIRT, BET, and the proposed method) used as pre-processes are given. In this context, the average obtained by the proposed method is measured as 0.946, 0.9049, 0.909, and 0.913 for Accuracy, F1 score, Precision, and Recall, respectively. In addition, the average performance of the studies performed with only FLIRT pre-processed samples was calculated as 0.7741, 0.8029, 0.8110, 0.7952, while the average of the studies performed with only BET pre-processed samples was calculated as 0.643, 0.7314, 0.7155, 0.7492 for the metrics as mentioned above, respectively.

In Fig. 3, the average of the regression performances obtained over the sub-problems with three different methods (FLIRT, BET, and the proposed method) used as pre-processes are given. In this context, the average obtained by the proposed method is measured as 0.9001, 0.8106, 0.1775, 0.0314, 0.1498, and 0.3851 for R, R², MAE, MedAE, MSE, and RMSE, respectively. In addition, the average performance of the studies performed with only FLIRT pre-processed samples was calculated as 0.7735, 0.5717, 0.3393, 0.0925, 0.3420,

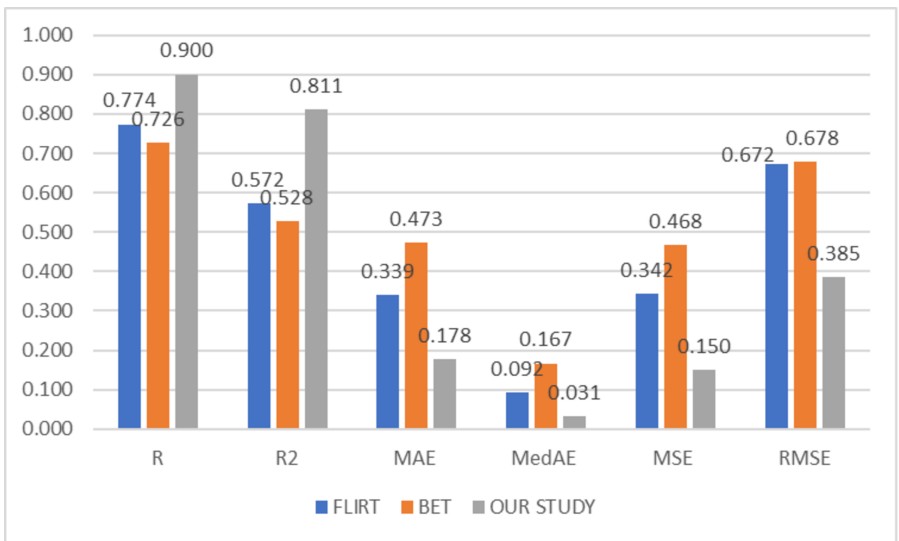

**Figure 3** **Average regression performance achieved by preprocessing techniques.**

0.6721, while the average of the studies performed with only BET pre-processed samples was calculated as 0.7263, 0.5281, 0.4734, 0.1666, 0.4681, 0.6775 for the metrics as mentioned above, respectively.

## DISCUSSION

This study deals with detecting Parkinson's disease with 2D and 3D CNN deep learning methods fed by T1w MR images and the problems of predicting disease severity.

In traditional methods, these pre-processes are carried out manually for each MRI individually. In these methods, the human factor directly manipulates the data. Since these processes require a long time to be done manually, one by one, the studies in this area generally have a small number of samples. In addition, the human factor that comes into play in the BET process adds subjectivity to this stage. The pre-processing part has been automated to eliminate the human factor at this stage and to process more samples.

After these procedures, MR images became suitable for the proposed 2D and 3D CNN deep learning methods. The architecture created for 2D CNN is inspired by the AlexNet architecture. Although there are different CNN architectures in the literature with more layers, the architecture used met the expectations regarding results and runtime performance. A relatively simple architecture is used for the 3D CNN architecture. It is anticipated that this situation affects the performance slightly, and using a more complicated architecture is not possible, given the number of samples used and sample sizes, for now.

The detection of Parkinson's disease corresponds to the classification, and the prediction of the disease severity conforms to the regression problem in artificial intelligence. In this context, the classification and regression results using 2D and 3D CNN are given in Tables 1 and 2, respectively. When both tables are examined, it can be observed that the results obtained are at a satisfactory level. More intense deep learning architectures can be used

when it is desired to increase this performance in future studies. Also, increasing the sample size would be a strong alternative and/or supportive approach. The use of $182 \times 218 \times 182$ original-size MR images and/or halved versions of these sizes, which cannot be used in this study due to the size of the data, also has a strong potential for performance enhancement.

To evaluate the effectiveness of the proposed hybrid approach, the dataset was initially processed using FLIRT and BET methods, as well as in its raw format, and then handled by artificial intelligence algorithms. During this analysis, noticeable improvement was not observed in the classification and regression models trained on the raw images. While FLIRT and BET pre-processing methods demonstrated positive outcomes, neither of them could match the level of results achieved by the proposed method in their respective sub-problems.

## CONCLUSIONS

In this study, 2D & 3D MRI analyses were performed using 2D and 3D convolutional neural networks for the detection of Parkinson's disease and estimation of its severity. The study utilized full-brain 3D MRI scans and median slices in the sagittal, coronal, and axial planes of these scans to understand complex patterns in all subcortical structures of the brain to detect and predict the severity of Parkinson's disease. To provide performance evaluation of CNN models, various evaluation metrics were used for classification, while another set of metrics mentioned was used for regression in 'Performance evaluation'. In classification, the best performance was obtained with the median slices technique. The results obtained were measured as 0.9620, 0.9452, 0.9407, and 0.9536 for Accuracy, F1 score, Precision, and Recall, respectively. In regression, the All Brain Downsized $\times$ 3 technique surpasses all other methods. The results obtained with the All Brain Downsized $\times$ 3 methods were measured as 0.9286, 0.8622, 0.1576, 0.0389, 0.1089, 0.33 for R, $R^2$, MAE, MedAE, MSE, and RMSE, respectively.

The result of the proposed study is motivating. However, a largely untouched area of work is involved in developing innovative architectures that can be utilized to detect Parkinson's disease and predict its severity using 2D and 3D CNN. Currently, the study has focused on whole-brain MRI scans and slices extracted from these scans in future research, it is aimed to develop a more efficient method for detecting Parkinson's Disease and predicting its severity with these structures by separating specific subcortical structures from the rest of the brain with neuroimaging methods.

### Funding
The authors received no funding for this work.

### Competing Interests
The authors declare there are no competing interests.

## Author Contributions

- Çağatay Berke Erdaş conceived and designed the experiments, performed the experiments, analyzed the data, performed the computation work, prepared figures and/or tables, authored or reviewed drafts of the article, and approved the final draft.
- Emre Sümer analyzed the data, prepared figures and/or tables, authored or reviewed drafts of the article, and approved the final draft.

## Data Availability

The data is available at Parkinson's Progression Markers Initiative (PPMI): https://www.ppmi-info.org/access-data-specimens/download-data.

## Supplemental Information

Supplemental information for this article can be found online at http://dx.doi.org/10.7717/peerj-cs.1485#supplemental-information.

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
