# Peer review of "A fully automated approach involving neuroimaging and deep learning for Parkinson’s disease detection and severity prediction"

_PeerJ Computer Science, doi:10.7717/peerj-cs.1485_

## Round 0.1 · original submission · Major Revisions

Based on the reviewer comments, I recommend a major revision of the manuscript. The author should improve the manuscript, taking carefully into account the comments of the reviewers in the reports, and resubmit the paper.

Reviewer 1 ·

Basic reporting

Technical writing is poor. Need improvement. There are typo mistakes. Related work presented in this paper is not sufficient. classification of section needs improvement. Result section not clear.

Experimental design

Research problem is not mentioned. Object of this paper is not clear. Very poor presentation. Proposed method/model must be supported in terms of pictorial form. It is difficult to visualise the model. Novel of work is not clear.

Validity of the findings

Difficult to understand the finding of this paper.

Additional comments

This paper lacks in many fronts.

·

Basic reporting

In this study, the authors have applied 2D and 3D CNN on T1-weighted MRIs for the detection of Parkinson’s disease and the prediction of disease severity. They used FLIRT image registration and BET non-brain tissue scraper for preprocessing the images. I would like to recommend for major revision keeping in mind the following points in the manuscript.

1. The authors should check the typographical and grammatical errors in the paper. English needs to be improved significantly.
2. Why have the authors considered the prediction of disease severity as a regression problem?
3. The authors should have compared their work with some other work available in the literature on the same dataset.
4. Is there any specific reason for using the FLIRT registration tool?
5. What was the criterion for using 0.5 as the threshold to eliminate unnecessary structures in the FSL’s BET method?

Experimental design

The experimental design is good.

Validity of the findings

The authors should have compared their results with some other work on the same dataset.

Additional comments

Article can be accepted after minor revisions and once clarifications sort in the comments are justified.

Reviewer 3 ·

Basic reporting

The article is addressing an important and timely issue. However, I have following comments to suggest the authors for improvements.

The novelty of a work is decided by the finding they have. In this work, authors have concluded that their work is promising, however, the supported result needs a comparison from the existing techniques.
This comparative evaluation is necessary.

Moreover, it seems that authors have unnecessarily added many self citation, for example, Accuracy (Erda_ et. al., 2016), F1 score (Aç1c1 et. al.,2019, Precision (÷lÁer & Erda_, 2022), and
Recall (G¸ney & Erda_, 2019) metrics were used, while the correlation coefficient (R) (Henseler,
Ringle & Sinkovics, 2009), R2 score (coefficient of determination) (Erda_, S¸mer & Kibaro, 2022), Mean Absolute Error (MAE) (Willmott & Matsuura, 2005), Median Absolute Error
(MedAE) (Chung et. al., 2008), Mean Squared Error (MSE) (Kˆksoy, 2005), and Root Mean
Squared Error (RMSE) (Chai & Draxler, 2014) were used to test the performance of the
regression methods developed for the prediction of the disease severity.

Accuracy and other metrics are much familiar and never required to be cited, or if citation required, it should be the first article addressing accuracy, doesn't it?

Experimental design

Experimental design seems fine. However, more clarity would surely enhance the paper presentation.

Validity of the findings

Please refer above comments.

---

## Round 0.2 · Major Revisions

Based on the referee reports, I recommend a major revision of the manuscript. The author should improve the manuscript, taking carefully into account the comments of the reviewers in the reports, and resubmit the paper.

·

Basic reporting

No comment

Experimental design

No comment

Validity of the findings

No Comment

Reviewer 3 ·

Basic reporting

To the comments previously suggested, authors have argued there is no work that blends "neuroimaging
and deep learning for Parkinson's disease detection ". What is the purpose of using both techniques for detection? obviously to enhance the result, right? For this purpose, authors can even compare with either of the technology, or they must do the comparison with neuroimaging separately and separately with DL techniques. So that the advantage of using blended (hybrid technique) can be seen.

Experimental design

ok

Validity of the findings

Refer to above comments.

Additional comments

no

---

## Round 0.3 · Minor Revisions

Based on the referee reports, I recommend a minor revision of the manuscript to address the language issues.

Reviewer 3 ·

Basic reporting

The authors have improved their work.

Experimental design

The comparison as asked, has been incorporated.

Validity of the findings

Fair.

Additional comments

However, still minor typos and English language errors exist, please revise thoroughly.

---

## Round 0.4 · accepted · Accept

I am accepting the manuscript for publication.